# Primary Root and Mesocotyl Elongation in Maize Seedlings: Two Organs with Antagonistic Growth below the Soil Surface

**DOI:** 10.3390/plants10071274

**Published:** 2021-06-23

**Authors:** Mery Nair Sáenz Rodríguez, Gladys Iliana Cassab

**Affiliations:** Departamento de Biología Molecular de Plantas, Instituto de Biotecnología, Universidad Nacional Autónoma de Mexico, Av. Universidad 2001, Col. Chamilpa, Morelos, Cuernavaca 62210, Mexico; gladys.cassab@ibt.unam.mx

**Keywords:** primary root, hydrotropism, drought, mesocotyl elongation, deep planting, early vigor

## Abstract

Maize illustrates one of the most complex cases of embryogenesis in higher plants that results in the development of early embryo with distinctive organs such as the mesocotyl, seminal and primary roots, coleoptile, and plumule. After seed germination, the elongation of root and mesocotyl follows opposite directions in response to specific tropisms (positive and negative gravitropism and hydrotropism). Tropisms represent the differential growth of an organ directed toward several stimuli. Although the life cycle of roots and mesocotyl takes place in darkness, their growth and functions are controlled by different mechanisms. Roots ramify through the soil following the direction of the gravity vector, spreading their tips into new territories looking for water; when water availability is low, the root hydrotropic response is triggered toward the zone with higher moisture. Nonetheless, there is a high range of hydrotropic curvatures (angles) in maize. The processes that control root hydrotropism and mesocotyl elongation remain unclear; however, they are influenced by genetic and environmental cues to guide their growth for optimizing early seedling vigor. Roots and mesocotyls are crucial for the establishment, growth, and development of the plant since both help to forage water in the soil. Mesocotyl elongation is associated with an ancient agriculture practice known as deep planting. This tradition takes advantage of residual soil humidity and continues to be used in semiarid regions of Mexico and USA. Due to the genetic diversity of maize, some lines have developed long mesocotyls capable of deep planting while others are unable to do it. Hence, the genetic and phenetic interaction of maize lines with a robust hydrotropic response and higher mesocotyl elongation in response to water scarcity in time of global heating might be used for developing more resilient maize plants.

## 1. Introduction

Maize (*Zea mays* L.) is a plant that belongs to Poaceae, the family of grasses. This is one of the most important crops in the world since represents a high percentage of total cereal production. The successful and continuous production of maize is key to food security [1,2,3]. With the current climate catastrophe scenario and the problems of food insecurity mostly due to soil erosion, drought, loss of arable land, and a population increase to approximately nine billion for 2050, it is projected that there must be an increase of 70% in global food production to mitigate hunger [4,5,6,7]. Thus, it is essential to develop different strategies to anticipate dire changes with the development of more resistant varieties of maize, capable of growing in diverse conditions and regions, where they can maintain high yields, as well as be nutritious [8,9,10,11]. Germination and early seedling growth are crucial for the establishment and seedling vigor of maize [12,13]. In these stages, the elongation of roots and the mesocotyl play a critical role beneath the soil surface that concludes with plant emergence. These agronomic characteristics have been more studied in rice, where it has been observed that higher yield often showed a direct association with seedling vigor and establishment. Accordingly, it has been used in rice breeding programs [13,14,15,16].

Since its inception, the approach to enhance crops has been to select superior genotypes from the identification of superior phenotypes. This selection has been used for years by phenotyping different traits of the aerial system (seed shape, size, or pigmentation), growth habits, and morphological markers (also “classical” or “visible”) of selection [8,10]. The traits of the organs that growth beneath soil surface have recently been analyzed as potential selection markers. In maize, the elongation of roots and the mesocotyl are traits of interest in the vegetative phase. These two organs with antagonist growth direction seem to determine the ability of seedlings to emerge under diverse conditions such as deep planting, drought, and cold or warmer soils [17,18,19,20].

Several regulatory mechanisms participate in the growth, development, and functions of roots and the mesocotyl; however, many of the processes that control the growth of these organs remain uncertain [21,22,23]. These are also influenced by genetic and environmental cues to guide their growth toward optimizing the establishment of the emerging young seedling, which will impact the full development of the mature plant [12,13].

Roots ramify through the soil following the direction of the gravity vector and spread their tips into new territories looking for water and nutrients; however, when water accessibility is decreased, roots develop a curvature toward the zone with higher moisture in the soil [24,25,26,27]. In maize hybrids, lines, and landraces, the root hydrotropic response trait varies widely from being robust (with root curvatures >40°) to weak (with root curvatures of <40°). This classification was established in accordance with observations in the test system of Eapen et al. (2017), which supports the great genetic diversity of maize [20,26].

Interestingly, the hybrids with a robust hydrotropic response were more resistant to drought and partial lateral irrigation compared to those with a weak hydrotropic response [26]. In turn, mesocotyl elongation is associated with an ancient agriculture practice known as deep planting, which is still utilized in the semiarid regions of Mexico and in the southwest of USA [28,29,30,31,32,33]. This practice takes advantage of residual soil humidity, thus enabling roots to absorb water for the developing seedling. However, not all maize lines have developed long mesocotyls capable of resisting deep planting [28,29,30,31,32]. Hence, the interaction between root hydrotropism and mesocotyl elongation in response to water deficit could be used to decipher the mechanisms that improved the resilience of one of the world’s most productive crop in a time of global heating.

Currently, it is urgent to invest in these traits in accelerated breeding programs for releasing new improved maize lines/cultivars/landraces with adequate yield under the forthcoming and irreversible changes such as higher temperature, unavailability of fresh water, pollution, and soil erosion. Farmers in developing countries, which will undoubtedly be the most affected by this crisis, will require access to elite germplasms with short breeding cycles, as well as resistance to drought, pathogens, and higher temperatures, especially since they practice low-input agriculture [34,35,36,37]. A detailed understanding of the genetic basis of traits such as mesocotyl elongation and the root hydrotropic response might help to accelerate genetic improvement through molecular marker-assisted selection (MAS) for the development of more drought-tolerant maize cultivars [35,36,37,38,39].

This review is focused on two antagonist growth traits that occur beneath the soil surface during the early phase of seedling growth and development that we propose should be incorporated in future breeding studies. Maize has several adaptations in response to a diverse array of abiotic stress conditions [39,40,41]. Many genes regulate many traits in order to adjust to several environmental conditions via the induction of various morphological, biochemical, and physiological responses [42,43]. In maize, pleiotropy (when one gene influences two or more seemingly unrelated phenotypic traits) appears to be minimal. Accordingly, a robust root hydrotropic response together with a high elongation of the mesocotyl is decisive in determining the ability of seedlings to emerge under diverse conditions, such as various cultivation practices, water limitation, and cold or warmer soils that could invigorate early vigor [1,44]. The ability to predict these traits from genotypes would be an extraordinary tool for breeders [4].

## 2. Deep Planting

Deep planting is an agriculture practice commonly employed at high altitudes or in drier habitats. Its implementation is principally based on the seed size and the capacity of the germinating seedling to take advantage of a major area of exploration and residual soil humidity [45,46,47,48]. The physical rationale for deep planting recommendation usually assumes a tradeoff of increased water and nutrient availability with the increased energy requirement for emergence as a function of depth [48,49,50]. This agriculture practice generally requires a mechanical disturbance to incorporate seeds deep into the soil, thus reducing the risk of either desiccation or adverse high-temperature effects near the surface. Furthermore, seeds may evade extreme difficulties at the soil surface, such as the possibility to be washed away by wind or water currents, as well as by predation. Shallow plants on the surface may adequately develop; nevertheless, they may expose their root system, which can be detrimental for the fitness of plants [47,49,50,51,52]. Fluctuations in light and diurnal temperature seem to regulate dormancy to ensure that seeds germinate at an appropriate depth for a given species. There is evidence that landraces domesticated by southwestern Native Americans during prehistoric times were selected for their ability to germinate up to >20 cm depth in the soil [28,29,30,31,32,50,51]. The central and eastern United States landraces seeds sown deep in the soil had mesocotyls with a medium degree of extensibility, while the Hopi maize resisted very deep planting in drier soils and had mesocotyls with the maximum degree of extensibility [29].

Maize grown in the world is commonly planted ~5 cm deep with the exception of some varieties of arid regions in the southwestern USA and parts of western Mexico where Native Americans or Mexicans cultivated their local varieties of maize at depths of 20 cm or more in an attempt to trigger germination with the moisture usually found in deeper soil horizons. Deep planting takes advantage of the early cool spring in climates where mature plants frequently encounter drought [28,29,30,31,32]. Successful germination and seedling establishment depend upon the placement of seeds in a favorable soil microcosm by reducing water loss and by redistributing or concentrating resources around the seedling that, together with the energy stored in the endosperm, will be sufficient for supporting the growth of the mesocotyl in order to force its way up to the surface [50,53,54,55]. Indeed, seedling roots have the ability to explore a widely area when they are cultivated by deep planting [45,46,47,48,49,50].

Most commercial maize varieties are not adapted for deep planting, and young seedlings are frequently handicapped when their seeds are sown below their optimum depth [1,45,54]. Therefore, researchers have identified the recommended planting depth ranges in relation to the type of soil, texture, pH, and moisture conditions for different crop species. However, soils are never uniform or level; thus, consistently maintaining deep planting becomes a difficult task [45,55].

New types of nondestructive sensor technologies for the development of in-field seed depth measurement have been used for sustainable and cost-effective maize cultivation [40,41,42,43,44,45,46,47,50]. The comparison between transmission seeding and drill seeding methods showed that two to three more transmission seeding rates were recommended in each planting cycle [45,52]. However, the evidence is still limited in relation to the effects of planting depth. These studies were carried out on a smaller scale in the field or in controlled environments, as shown in Figure 1. An in vitro test system allows simultaneously measuring two traits (the elongation of the mesocotyl and the hydrotropic response of the root) under different stresses conditions (deep sowing and low humidity gradient) while keeping numerous abiotic factors under control such as light, temperature, and water availability [48]. It is necessary to evaluate the effect of depth on a large scale with studies that quantify the specific depth distribution after planting [28,29,52].

Deep planting also depends upon the topography of the soil and the economic considerations regarding the type of planting equipment and number of farmers available [28,29,47,55].

Deep planting is an alternative agricultural practice that has a major influence on plant germination and final crop yield; hence, the identification of varieties of maize tolerant toward deep planting and with a robust hydrotropic response should be of uttermost importance for its application in sustainable agriculture in times of imminent drought due to the climate crisis [28,29,55,56,57].

## 3. Maize (*Zea mays* L.)

### 3.1. Growth and Physiology

Maize kernels present a persistent endosperm and an embryo already developed with primary and seminal roots, mesocotyl, scutellum, coleorhiza, and coleoptile [3,8,22]. The embryo proper contains the root meristem (RAM) and the shoot meristem (SAM), relevant during the vegetative and reproductive phases for the development of the plant; these consist of a pool of stem cells that divide to give rise to more daughter cells that undergo cell division and differentiation throughout the entire life of the plant (Figure 2). The roots and mesocotyl continue to grow using a combination of cell division in the meristems and cell elongation in subapical regions named elongation zones, which are highly controlled by several mechanisms [16,58,59,60,61].

Tropisms are defined as the directional growth of different organs of plants due to the gravity vector (intrinsic on Earth) and to several environmental stimuli, such as light, water, obstacles, nutrients, oxygen, and temperature. The mesocotyl typically grows against the gravity vector (negative gravitropism) to push the coleoptile with the enclosed plumule (first leaf) in order to move through the soil and reach the light (positive phototropism), whereas roots grow downward into the soil following the direction of the gravity vector (positive gravitropism). Then, new and old roots spread their tips into new territories surveying for water and nutrients; when water is scarce, roots develop a hydrotropic response toward the zone with higher water accessibility [27,62,63,64,65].

The root system architecture of maize (*Zea mays* L.) has a complex organization. It contains embryonic and postembryonic root types, which are formed during different stages of development [60,66,67,68,69,70,71]. These are regulated by genetic programs, by interactions with the rhizosphere, and by adapting to changing environmental cues [59,72,73,74]. The primary root dominates the early seedling phases, which are the focus of this review.

Maize germination occurs in the dark when the seed contains at least 30% (*v*/*w*) moisture [61,74]. Initially, the primary root (PR) is formed at the basal pole from the embryonic radicle, and a variable number of seminal roots (SR) are laid down at the scutellar node. The ontogenesis and the role that SRs play in determining a good performance of the young seedling are still uncertain [60,74,75,76,77]. The PR elongates rapidly and, as it grows, forms many new lateral roots (LRs). The LRs initiate from a pericycle cell or sometimes from endodermal cells of the differentiation zone of the PR, and they increase the absorption surface of the root system, later becoming essential for water and nutrient uptake in young seedlings [71,78,79,80,81,82,83].

The PR may persist throughout the plant’s life, although it frequently ceases growing and branching, and it has relatively little functional importance after seedling establishment [65,66,67,68,84]. The PR commonly shows an orthogravitropic response (vertical to the gravity vector), while the SRs and LRs display diagravitropism (horizontal to gravity vector) or plagiogravitropism (oblique to the gravity vector). When the maize plant is in vegetative stage I, further lateral organs emerge from the primary organs in a stereotypical manner although they quickly realign to follow the gravity vector [62,63,64,65,71].

The presence of various functionally diverse cell types in the primary root determines its radial organization in a transverse orientation [76,78,79,80]. The root vascular system (xylem and phloem) is constituted by a central cylinder that contains the elements necessary for water and nutrient transport, where its outermost cell layer is a single anatomically distinct layer of thin-walled pericycle cells, followed by the cortex and the cuticle-less epidermis [71,83,84,85,86]. Longitudinally, maize roots can be divided into the root cap, followed by the RAM, the elongation zone, and the differentiation zone characterized by the presence of root hairs, which are formed on both embryonic and postembryonic roots and have crucial roles in water and nutrient uptake [80,82,83,84,85,86,87]. The postembryonic root system is composed of shoot-borne roots that are formed at consecutive shoot nodes and LRs that are initiated in the pericycle of all roots. Nodal maize roots can be subdivided into crown and brace roots according to whether they initiated from underground or aboveground nodes, respectively. They are crucial for water absorption and anchorage during the further development stages of the maize plant [69,71].

The mesocotyl can be seen as a bridge that connects the basal part of the seedling with the coleoptilar node and shoot, and both the mesocotyl and coleoptile push the shoot tip above the soil surface during seed germination [88,89]. In maize, this structure was defined by a group of researchers as the first portion of the shoot, while others considered it as a modified root with a cuticularized epidermis since its physiology is quite similar to that of roots [28,29,30,31,32,90]. The morphology and the anatomy of the maize mesocotyl are described as having several zones. In transversal slides, observed from outside to inside, these are the epidermis, cortex, and central vascular stele; in longitudinal slides, the first zone observed is next to the coleoptilar node (approximately 1.0 mm), including meristematic cells, the second zone (3.0 mm below) contains rapidly growing cells, and the third zone (5.0 mm below) near to the escutellar node presents mature cells (Figure 3) [61]. The mesocotyl also uniquely responds to environmental cues and to stress conditions, whereby some maize lines develop fine roots capable of extensive growth [91]. The dominant factor implicated in mesocotyl elongation is somatic polyploidy referred to as endopolyploidization [89,92]. Polyploidy is defined as an increase in genome DNA content. Somatic polyploidy seems to be a mechanism that produces large cells. This is utilized in some developmental contexts in which fewer and larger cells have functional advantages over a similar bulk of an increased number of smaller cells. Polyploidy likewise increases gene expression and/or metabolism [89,93]. It remains to be determined whether the cells of the mesocotyl become larger after germination, and whether this phenotype may ultimately regulate its elongation and function.

The effects of anatomical and architectural changes in these two organs (root and mesocotyl) in response to various types of stress are also of interest for crop improvement.

### 3.2. Regulatory Mechanisms

The transition from the skotomorphogenic or etiolation period to the photomorphogenic period is primarily regulated by the lengthening of the roots and the mesocotyl; nevertheless, different regulatory mechanisms control their growth and development. Seedling development outlines the future plant body with preformed embryonic organs, including the mesocotyl and the roots. The two main factors that regulate seedling development are (a) light (phototropism), which has been widely studied, and (b) the intervention of growth regulators that influence various physiological process at low concentrations (hormones such as auxin, gibberellin, and ethylene) [28,29,30,31,32]. Both factors can act alone and be integrated into several processes during the establishment of maize seedlings (bounded by the red dotted lines; refer to Figure 4). During this time, the primary root and mesocotyl simultaneously elongate in dark conditions, using the seed energy and with minor water consumption compared to that in other phenological cycle phases such as flowering, as described below (Figure 4) [94,95,96].

#### 3.2.1. Light

The patterns of development under light are different from those in the dark in relation to gene expression, cellular and subcellular differentiation, and organ morphology [22,52,94]. The light penetrates several cm through the soil, creating gradients of light quality and quantity [52,94,95,96,97,98,99]. Therefore, it is also necessary to evaluate the depth of sowing according to the practice of the crop. Under current cultivation practices, maize seeds are sown within a few centimeters (~5 cm) of the soil surface, with the exception of some cultivars that have the ability for germinate when sown at 20 cm depth, utilized in deep planting [96,97,98,99,100].

In darkness, the energy required for the elongation of the roots and mesocotyl is supplied by the endosperm reserves [58,59]. The elongation of these structures is primarily guided by two main stimuli, the gravity vector (gravitropism) and obstacles (thigmotropism), according to the postulate made by Knight in 1806, who indicated that a plant’s perception of gravity might modulate its ability to direct shoots to grow upward and guide roots downward, and who introduced the concept of tropism [66]. Years later, Darwin (1880) made numerous physiological contributions by documenting a wide array of tropic responses and identifying regions specialized for the perception of the different stimuli [63,95,96].

Nearing the soil surface, the seedling perceives both light and the dark–light transition, which affects organ architecture and growth rate during the first stages of seedling establishment. In particular, a young seedling buried under the soil surface showed rapid extension growth of hypogean organs and a reduction in or even inhibition of mesocotyl elongation [101,102]. After germination, the shoot apex sheathed by the coleoptile is pushed through the soil by the elongating mesocotyl. The rapid upward movement of the shoot apex is facilitated by reducing root formation and leaf expansion, while expending the last amount of energy derived from the seed reserves [103]. As the emerging maize seedling perceives light, there is a decrease in mesocotyl elongation, an induction of root growth, and an expansion of leaves, along with numerous profound changes that allow the initiation of the photoautotrophic phase of the plant [22,94,95,96,97,98].

In maize seedlings, the processes associated with the transition from dark to light conditions have barely been studied in comparison with those of *Arabidopsis thaliana* (Dicotyledoneae model). In *Arabidopsis*, these have been ascribed to global transcriptional changes mediated by relocalization, phosphorylation, and degradation of light response regulators [22]. It is likely that many of these early signaling events are similar in grasses; however, they need to be determined fully in monocots [71,104].

Photosynthesis requires active radiation between 400 and 700 nm (white light, W), and the maximum absorption spectra of plant pigments are as follows: chlorophylls (430–455 nm) and carotenoids (400–500 nm). Much of the green and most of the far-red spectra are either transmitted through green tissues or reflected by the plant and, thus, have the potential to serve as cues in sensing the foliar environment [71,98]. However, higher plants have several photoreceptors, such as phytochromes, cryptochromes, phototropins, and UV-B photoreceptors, which allow the developing seedling to not only monitor the quality and flux of incident light, but also control its growth and further development [22,105]. Plants typically perceive wavelength ranges such as UV-B (280–320 nm), UV-A (320–380 nm), blue (380–495 nm), green (495–570 nm), yellow/orange (570–620 nm), red (620–690 nm), and far-red (690–800 nm) [96,106]. Light may be involved in many more developmental processes during the entire life of the plant, which are still unknown in relation to its broad spectrum of perception, thus necessitating further studies.

Seedling growth just after germination is profoundly influenced by light in two important ways: (1) by phototropism, which has been extensively reviewed and is not discussed in this review, and (2) by light inhibition of organs, which is equally important in saving energy during growth [71,107]. As mentioned earlier, the elongation of the roots and mesocotyl is stimulated in the dark before emergence, indicating that both organs first display a negative phototropic response. However, growth of both the roots and the mesocotyl is slowed down by light, perhaps by the presence of chlorophyll in the early stages of seedling development [31,106].

Root growth and mesocotyl elongation have shown inhibitory responses to red light, far-red light, UV light, and blue light dependent on the intensity and time of exposure [96,105,106,107,108,109,110,111]. Blue light induces the coleoptile to supply to the mesocotyl a substance that slows its rate of elongation. This inference was based upon (1) the presence of a factor in the coleoptile exposed to blue light, (2) the presence of pigments in the coleoptile that absorbed blue light, and (3) the apparent absence of such pigments in the mesocotyl [106,110,111,112]. Cell-wall extensibility also inhibits mesocotyl elongation in the presence of light [112].

The first photomorphogenic mutant of maize *elm1 (elongated mesocotyl 1)* was identified and characterized in a sand bench screen as a pale, green plant with an elongated mesocotyl. The *elm1* encodes an enzyme with phytochromobilin synthase activity, which disrupts the synthesis of the chromophore of phytochrome and makes it very deficient in phytochrome responses to both red and far-red irradiation. In this mutant, the phototropic response of mesocotyl elongation has been studied extensively; nonetheless, relatively few investigations have focused on root phototropism during the pre-emergence phase [107,108,109,110,111,112]. In maize, little is known about the molecular components of the light signal transduction pathways. However, the critical role of light is recognized in the regulation of the root and mesocotyl elongation traits, indicating that the manipulation of light responses in these organs may prove fruitful in enhancing seedling vigor and should be examined further [98,104,105,106,107,108,109,110,111].

Additionally, polyamine oxidase (PAO) increases its activity after light exposure in maize mesocotyl, which results in the hardening of the cell wall and inhibition of mesocotyl elongation [95,112].

#### 3.2.2. Hormones

Plant hormones play a crucial role in controlling the way in which plants grow and develop. These regulate the speed and timing of growth of the individual organs and cells and integrate them to generate the basic body plan that we recognize as a plant (Figure 5) [95,96,97,113].

Indole-3-acetic acid (IAA) is the main auxin in plants. Auxin stimulates elongation of the roots and mesocotyl and is also implicated in the control of the tropic responses to gravity and light [23,114,115]. It has been suggested that, during mesocotyl elongation, light regulates auxin supply from the coleoptile. Auxin induces mesocotyl growth and suppresses it via inhibition of its transport [95,116,117]. The mechanisms of regulation of gene expression, transport, and function of auxins in the mesocotyl are unclear [117,118]. However, Dong et al. (2013) described the Zm*LA1* gene that encodes a *LAZY1* ortholog in rice and *Arabidopsis*, which plays an important role in the regulation of polar auxin transport and auxin signaling in maize, influenced by light. Thus, mesocotyl elongation in *la1* mutants was accelerated in dark but not in light conditions [119,120].

Abscisic acid (ABA) promotes the elongation of roots and the mesocotyl by increasing cell division activity of the meristem [95,121]. ABA also regulates shoot growth, which suggests that the capacity for inhibiting mesocotyl expansion increases as cells are displaced away from the meristematic regions of both roots and shoots [122,123]. Previous work has indicated that accumulation of ABA acts differentially to maintain elongation of the primary root and inhibit elongation of the mesocotyl of maize (*Zea mays* L.) seedlings at low water potentials [122,123,124].

Gibberellins (GA) are essential regulators of many aspects of plant development including organ elongation [95,125]. This hormone increases the amount of osmotic solutes, such as sugars in shoots of various plants, thereby stimulating elongation growth of shoots and cell-wall-bound invertase (INCW) activity by suppressing INCW activity via downregulation of its gene expression in both organs. This is enhanced in the mesocotyl. This enzyme is ionically bound to the cell wall and catalyzes the hydrolysis of sucrose to glucose and fructose, and it is involved in the phloem unloading process of sucrose in sink tissues with sugar accumulation. Cellular osmotic properties are an important factor regulating the rate of cell expansion. It is generally believed that the amount of osmotic solutes in cells determines the force of water uptake, thereby regulating the rate of cell growth [126]. However, Soga et al. (2021) in their studies showed the decrease in sugar content in coleoptiles and mesocotyls and inhibited the growth of both organs when etiolated maize seedlings were subjected to light irradiation. Thus, the phenomena induced by light were opposite to those induced by gibberellins [127,128,129]. Hoson (1999) proposed that light acts as a gravity-substituting factor in growth regulation, because many mechanisms for growth regulation by light and gravity are common. The GA response of maize mesocotyl is cultivar-dependent. However, all maize seedlings from different inbred lines are insensitive to GA under shallow growth conditions [130,131,132].

Ethylene (E) is a hormone produced from methionine, which inhibits the elongation of maize roots and the mesocotyl in darkness [133,134,135]. This effect on the mesocotyl is rescued by adding carbon dioxide (CO_2_) [134]. Ethylene production is highly regulated by several enzymes encoded by multigene families, and it is an essential mediator of plant responses to environmental stimuli [135]. In maize roots, E upregulates the expression of cell-wall-associated proteins including xyloglucan endo-transglycosylase (Zm*XET*), which swells and assembles plant cell walls [136,137,138]. Furthermore, ethylene stimulates elongation of the mesocotyl under red light conditions and induces lateral expansion [95,139].

Brassinosteroids (BR) are part of a family of plant-specific endogenous and polyhydroxylated steroid hormones [140,141,142,143]. BRs play a crucial role in many physiological processes throughout the lifespan of the plant, from germination to seed production, and are involved in mediating plant responses to stress [142,143,144]. The biosynthesis and signaling pathways in maize have not been well established, but their defects seriously affect plant physiology during photomorphogenesis in the dark [143,144,145]. BR and GA promote vigor and growth of seedlings [131].

Jasmonates (JA) are oxidized lipids, collectively known as oxylipins [146]. These inhibit many plant processes such as seed germination and seedling growth. The molecular mechanism of this process underlying plant growth remains to be elucidated in mesocotyl and root elongation [147]. In *Arabidopsis*, the antagonist activity of JA against the activities of IAA and GA, is related to the action of different signaling elements such as auxin-signaling *AXR1* (*AUXIN RESISTANT 1*), whereby *axr1* mutants displayed a variety of morphological defects consistent with a reduction in auxin sensitivity, and GA-signaling protein DELLA, which acts as a repressor of GA activity [147,148,149]. On the other hand, the double-mutant for two oxo-phytodienoate reductase genes *OPR7* and *OPR8* (*opr7opr814*) in maize had a dramatically low content of JA, E, ABA, and cytokinin (KIT) and showed developmental defects such as an unusual ear [147,148,149].

Strigolactones (SLs) are butenolide compounds that are involved in various developmental processes and environmental responses, including root system architecture formation, secondary growth of the cambium, inhibition of mesocotyl elongation by controlling cell division in the dark, and drought tolerance in rice (*Oriza sativa* L.) [95,150,151].

Active KIT consist of an adenine (Ade) moiety with an N6-substituted isoprene chain, isoprene derivative, or an aromatic ring [95,151]. Cytokinins inhibit LR formation, influence the root architecture, and seem to antagonize auxin action in the root meristem by promoting cell differentiation along the primary root axis [152].

The regulatory mechanisms described above show the complex interplay between light and hormone signaling pathways; thus, a thorough characterization of these networks in maize will most probably provide many potential targets for agronomic improvement [95,109,113].

### 3.3. Function

Root and mesocotyl elongation play important roles in different processes that end up with the emergence of the seedling, which might be considered as an index of seedling vigor. Both are sensitive organs that perceive and assess soil conditions, as well as chemical and biological factors in their environment for promoting seedling fitness. Changes in these organs have occurred as a consequence of domestication, breeding, and climate change, leading to contrasting spatial arrangements [61,153]. The particular functions of the roots and mesocotyl must integrate the mechanisms that regulate and coordinate at some level the local and systemic responses during seedling establishment. The functions of the maize root system prior to the emergence period have been poorly studied. These roots are critical for anchoring the plant in the soil and for efficient uptake of water and mineral nutrients, thus playing a key role in vigor [32,71,124]. Moreover, they contribute to widening the volume of soil explored by increasing the extension capacity of the root system and compensating for root death [46,71,132].

The architecture of root systems describes the spatial distribution of all type of roots, which are limited by the physical, chemical, and biological properties of the soil [45,50,71]. The root system has been the target of modern breeding programs to develop varieties tolerant toward drought or nutrient deficiencies. Unfortunately, it is very difficult to analyze the root system of many samples in field conditions without disrupting them [32,53,92].

The mesocotyl’s primarily function is to assure the establishment of the root system at a suitable distance below the soil surface, and it is also crucial for the emergence of the coleptile, which is the protective sheath enclosing the young leaves above ground. Additionally, it supports shoot-borne crown roots through its cortical aerenchyma. Thus, mesocotyl elongation and root maize development are strongly interrelated [30,31,154].

The mesocotyl is the first internode of the maize plant and is the organ that compensates for deep planting of the kernel. The mesocotyl continues to grow until the coleoptile emerges, after which it stops growing [31,55,89]. A healthy mesocotyl is extremely important because it transports nutrients from the endosperm of the grain to the developing seedling. The plant primarily depends on the endosperm content for its nutrients and energy until the nodal roots are developed [56]. Hence, seedlings may be stunted or even die if their nodal roots do not develop before the kernel reserves are exhausted. Similarly, if the primary leaf emerges underground, it is likely that the seedling perishes [56,155,156,157].

The results of Cheeseman et al. (1983) indicated that the mesocotyl of maize seedlings is crucial in the control of sodium transport to the leaves, while maize roots showed marked selectivity for potassium over sodium during the vegetative phase of development [90,119,158]. Thus, an effective exclusion of cations (Na^+^) by the maize mesocotyl can be utilized as a potential strategy for salt tolerance of the shoot.

## 4. The Root Hydrotropic Response and Mesocotyl Elongation for the Genetic Improvement of Crops

The strong effects of climate change or, as recently described, climate catastrophe on the yield of different crops in the last few years have aroused common interests among plant scientists and breeders for examining underground traits that could be used in the genetic selection of more resilient plants exposed to higher temperatures and longer droughts [32,33,34,35,36,37,38,39,40,159]. Despite the difficulty in simultaneously examining the roots and mesocotyl under realistic conditions, genomic analyses of these organs in maize have increased substantially, resulting in the dissection of many features of agricultural importance [19,32,33,34,35,36,37,38,39,40]. The availability of the B73 maize reference genome, the various accessions that have been extensively collected in recent years for quantitative purposes, and the complexity of the genome, which has demonstrated the presence of several genotypes and whose various characteristics provide abundant genetic material, make maize a noteworthy study model [160].

Recent advances in DNA sequencing technology, including major access to next-generation sequencing (NGS), have enabled powerful tools to be used for understanding the genetic basis of these variations [9,160,161,162]. Thus, we can apply various methodologies that include breeding and selection in the field combined with QTL mapping, genome-wide association studies (GWAS), comparative transcriptomes, and precise high-throughput phenotyping of diverse complex traits for selecting superior varieties of maize [17,33,34,35]. However, identifying the causal gene or variant is still challenging [39,40,41,42].

### 4.1. Quantitative Trait Locus Mapping (QTL)

QTL analysis using different molecular techniques has permitted the identification of genes in chromosome regions and their association with traits at diverse stages of development, under diverse biotic and abiotic conditions. This has also enabled establishing the number of individuals and generations required for improvement (introgression or segregation of characters into the population). The main disadvantage of QTL mapping is the great influence of environmental factors [32,33,34,35]. However, QTL analyses of these two traits have shown relatively small effects or failed in their detection under some genetic backgrounds, while their localization has not been consistent across studies, which is necessary for any future map-based cloning and marker-associated selection [160,161,162,163]. Zhang et al. (2012) and Pace et al. (2015) demonstrated the main role of mesocotyl elongation in deep seedling tolerance and the beneficial influence of a larger root system for the maize plant. These results were dependent of the target environment and their test maize population, where the traits seem to be controlled by several loci, and they also revealed considerable overlapping among the QTLs for different traits of the root system [138,139]. Root elongation showed a correlation with other traits such as branching of the root system (e.g., number and weight of seminal roots), while mesocotyl elongation showed a correlation with coleoptile elongation, as well as early vigor (Table 1) [103,162,163,164].

### 4.2. Genome-Wide Association Studies (GWAS)

In recent years, GWAS have become a useful adjunct to classical genetic mapping of quantitative traits in plants (Huang et al., 2013). This powerful statistical approach to identify the genetic factors underlying intraspecific phenotypic variations has allowed us to acquire variation data on a large number of accessions characterized by various traits. The GWAS approach has been successfully applied for important aerial and underground characteristics of important crops [14,163,164,165,166]. In maize, GWAS have identified markers associated with the development of root traits under various conditions, whereas there are no GWAS reports of mesocotyl elongation; however, there have been some studies in other grasses such as rice [161,162,163,164,165,167,168,169,170,171]. Analysis and interpretation of the cause–effect relationship among different traits will be useful for understanding how maize avoids drought under diverse enviroments [32,33,34,35,166,172,173,174].

### 4.3. RNA Sequencing (RNA-Seq)

RNA-seq is an NGS assay, whose low cost, high throughput, and high sensitivity have conveniently aided in carrying out transcriptome analyses of maize [175]. These assays have allowed obtaining complete transcripts from the RNA of maize root tissue or from individual cells, as well as the mesocotyl of other grasses. The information derived from RNA-seq analysis might aid in the identification of the developmental processes that occur throughout the life cycle of a plant [175,176,177]. Consequently, the knowledge about gene expression networks modulating these two traits and their abiotic stress adaptation could be used for genetic prediction, and this might be exploited not only for direct selection in breeding programs but also for identifying genotypes of interest, which may have rare alleles affecting quantitative traits of interest that are difficult to measure [164,177,178,179].

At present, the SNPs related to the traits and the candidate genes through which the identified genetic variants exert their effects on the traits remain largely unknown. Therefore, it is important for researchers to integrate multi-omics studies into linkage and association mapping to close the knowledge gap [32]. An understanding of the molecular regulatory mechanisms of traits in response to stress conditions will be useful for maize breeding [35,36,37,38,39,40,41,42,155,156,157,158,159]

## 5. Conclusions

Deep planting should be a more common practice for developing sustainable agriculture in arid and semiarid areas of the Earth in the current climate crisis due to human activity.New genetic and molecular tools should be used to examine the role of hormones and their interplay in controlling root hydrotropism and mesocotyl elongation. This characterization in maize will provide many potential targets for agronomic improvement.The genetic diversity of native maize landraces should be exploited for the selection of those which show a robust hydrotropic response and higher elongation rate of the mesocotyl for deep planting in poor areas without high-input agriculture and which are prone to longer droughts and higher temperatures.Maize improvement by genetic selection should use the information derived from genomic and transcriptomic analyses of root hydrotropism and mesocotyl elongation traits during seedling establishment in relation to early vigor.

## Figures and Tables

**Figure 1 plants-10-01274-f001:**
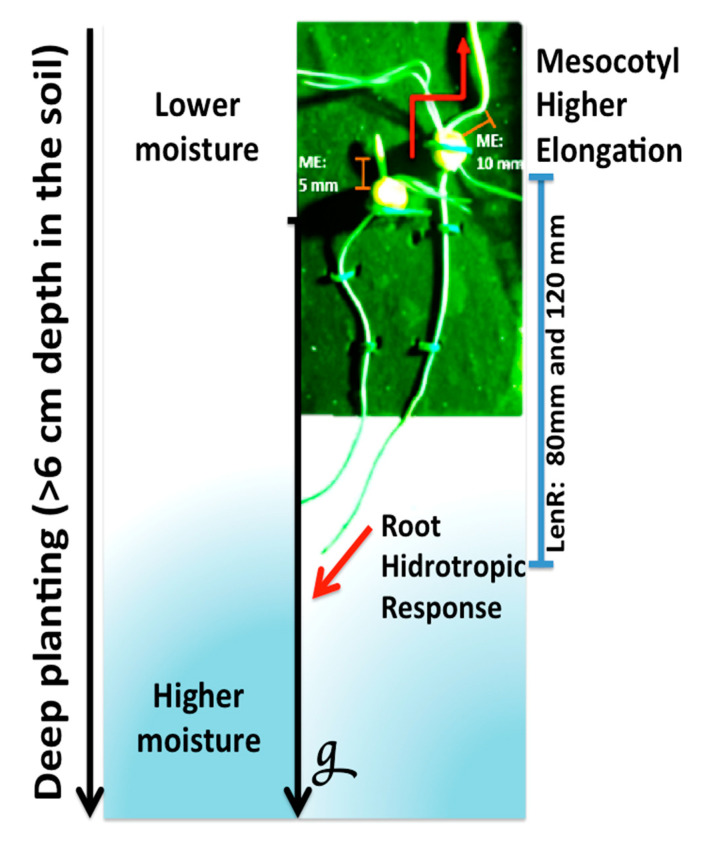
Establishment of maize seedlings when utilizing deep planting as an agricultural practice. Elongation of the roots (positive gravitropic and hydrotropic response) and mesocotyl allows the exploration of a wide area of soil by helping to forage water, which ultimately favors early seedling vigor (g: gravity; LenR: root length of 80 and 120 mm, ME: mesocotyl elongation of 5 mm and 10 mm).

**Figure 2 plants-10-01274-f002:**
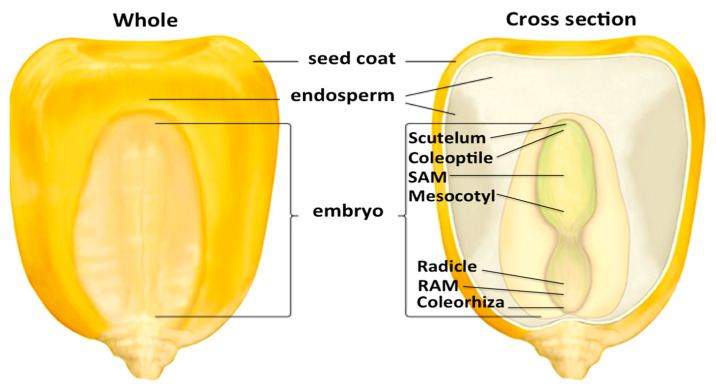
Maize kernel structure. The mature maize kernel consists of multiple tissues and organs within the embryo and endosperm, in addition to maternally derived structures. © 2013 Encyclopedia Britannica, Inc., modified image. (RAM: root apical meristem; SAM: shoot apical meristem).

**Figure 3 plants-10-01274-f003:**
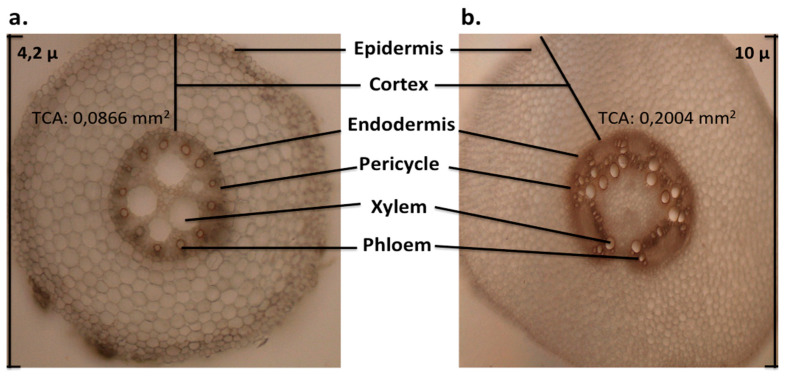
Cross-sections of primary roots of maize breed B73 at 7 days post germination. (**a**) Anatomy of primary root (10×); (**b**) anatomy of mesocotyl (10×). The lines indicate major structures. (TCA: total cortex area, RootScan®).

**Figure 4 plants-10-01274-f004:**
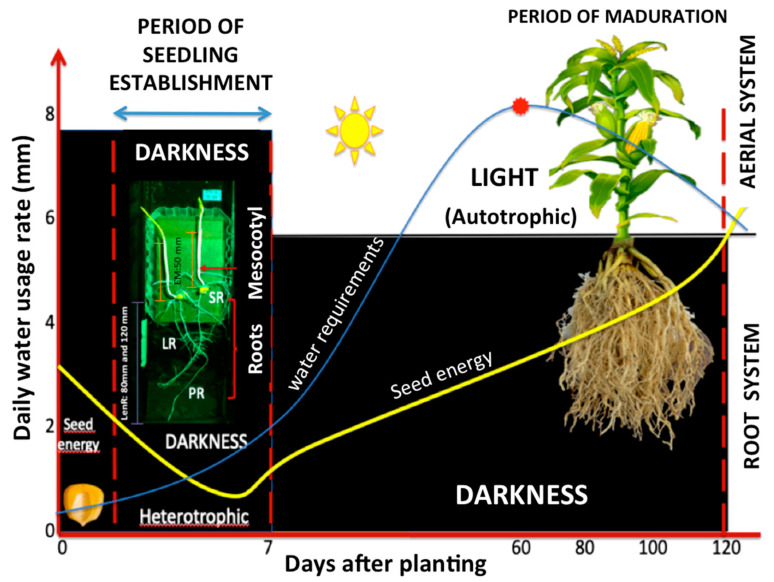
Establishment of maize seedlings. The transition from the skotomorphogenic or etiolation (heterotrophic growth) period to the photomorphogenic period (phototropic growth) is primarily regulated by growth of both the roots (PR, LR, and SR) and the mesocotyl. Growth of the roots and mesocotyl is controlled by several regulatory mechanisms, mainly tropisms and hormone signaling (M: mesocotyl, PR: primary root, SR: seminal root, LR: lateral root; LenR: root length of 80 and 120 mm; ME: mesocotyl elongation of 50 mm). Blue line: water requirements throughout phenological cycle of maize; yellow line: energy consumption of the seed.

**Figure 5 plants-10-01274-f005:**
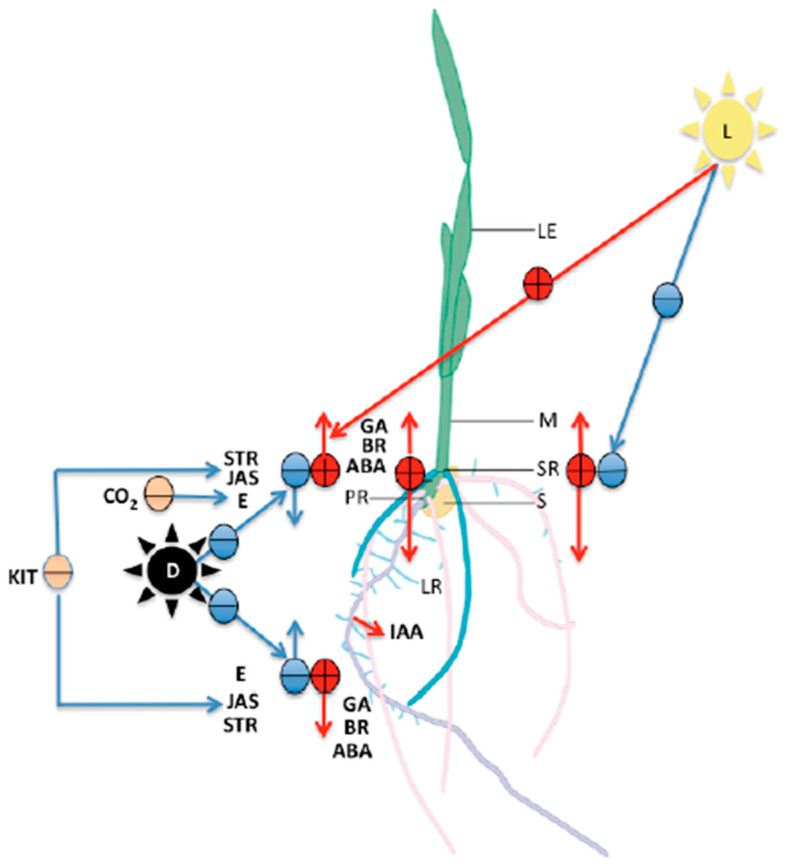
Diagram that shows the main effects of hormones on seedlings traits. (IAA: indole-3-acetic acid, ABA: abscisic acid, GA: gibberellins, E: ethylene, STR: strigolactones, JAS: jasmonates, BR: brassinosteroids, KIT: cytokinins). Their function in relation to growth and development of both the roots (PR: primary root, SR: seminal root, LR: lateral roots) and the mesocotyl (M). (S: seed, LE: leaves, CO_2_: carbon dioxide, D: dark conditions, L: light). Red circles with lines indicate a positive stimulus and its direction; blue circles with lines indicate a negative stimulus and its direction; nude circles indicate a negative stimulus.

**Table 1 plants-10-01274-t001:** List of QTLs related to drought and deep seedling tolerance identified by linkage mapping and association mapping in maize population.

Method	Abiotic Stress	Traits	Marker	Locus/ QTLs	Candidate Gene	References	Maize Populations
**Linkage mapping**	Drought tolerance	Root traits *	RFLPs	13 QTLs	None	Ruta et al., 2010	Ac7643xAc7729/TZSRW
			SNPs	364 QTLs	None	Li et al., 2018	DH1MxT877
	Deep seeding tolerance	Mesocotyl elongation	SSR	25 QTLs	None	Zhang et al., 2012	Inbred line 3681-4 and X178
				11 QTLs	None	Liu et al., 2017	lines of IBM Syn10 DH population and their parents B73 and Mo17
**GWAS**	Drought tolerance	Root traits *	SNPs		GRMZM2G153722	Paece et al., 2015	Inbred lines populations
			SNPs		GRMZM2G148106	Zhang et al., 2019	Inbred lines populations
			SNPs		GRMZM2G136364	Guo et al., 2020	Inbred lines populations

***** Root traits: primary root length, seminal root length, axial root length, lateral root length, crown root, and number of seminal roots.

## Data Availability

Not applicable.

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
