# Peer review of "Primary Root and Mesocotyl Elongation in Maize Seedlings: Two Organs with Antagonistic Growth below the Soil Surface"

_plants, 2021, doi:10.3390/plants10071274_

Round 1

Reviewer 1 Report

   Maize illustrates one of the most complex embryogenesis in higher plants that results in the development of early embryo with distinctive organs such as the mesocotyl, seminal and pri-mary roots, coleoptile, and plumule. After seed germination, the elongation of root and mesocotyl follows opposite direction in response to specific tropisms (positive and negative gravitropism and hydrotropism). Tropisms are differential growth of an organ that is directed towards several stimuli. Although the life cycle of roots and mesocotyl and their growth and functions are controlled by different mechanisms were well known, however, the processes that control root hydrotropism and mesocotyl elongation is still unclear,In this review paper, the authors summarized the process of primary root and mesocotyl elongation in maize seedlings in details.

   In general, this manuscript is well written and the data and figures are well-presented and the review will bring us the wide knowledge about the processes that control root hydrotropism and mesocotyl elongation in maize. Thus, from my point of view, this manuscript is suitable to be accepted to be published in this journal.

One minor comment:there is no reference cited from 2021.

Author Response

Dear Reviewer 1

Thank you! We found your comments very beneficial and have revised them accordingly.

#Manuscript ID: plants-1198014 Type of manuscript: Review Title: Primary root and mesocotyl elongation in maize seedlings: Two organs with antagonistic growth below the soil surface

  • Comment 1: There is no reference cited from 2021.

Response: Despite our extensive search for information over time about primary root and elongation of mesocotyl in maize seedlings. There is much information from individual traits but very few studies on both traits in maize, and also during pre-emergence period. This is one more motivation for our review and future research on this particular topic.

We have added one reference cited in 2021, in the hormone 3.2.2.section.

Changes: One reference cited from 2021 was included in the hormone 3.2.2 section.

Reviewer 2 Report

Dear authors, your manuscript is very interesting in the field.  The manuscript si vell organized and the essential facts are well explained. I found only three things that might need to be fixed and that is:

  1. in line 118 (Zea mays L.)
  2. in line 134 point (dot) at the end of the sentence and
  3. in Figure 2 whether the hybrid B73 or what?

After you fix the above your manuscript is ready for publishing. Thank you!

Author Response

Dear Reviewer 2

Thank you! We found your comments very helpful and have revised them accordingly

# Manuscript ID: plants-1198014 Type of manuscript: Review Title: Primary root and mesocotyl elongation in maize seedlings: Two organs with antagonistic growth below the soil surface

  • Comment 1: in line 118 (Zea mays)

Response: We added the correct terminology; this was a mistake of edition.

  • Comment 2: in line 134 point (dot) at the end of the sentence.

Response: We had overlooked this mistake, and we had already rectified it. We also had looked for other possible grammar mistakes.

  • Comment 3: in Figure 2 whether the hybrid B73 or what?

Response: We did the modification as suggested in Figure 2, in the 3.1. section. This figure changed its number, it is now Figure 3.

We did all the corrections according with your nomenclature, grammar and style observations.

Reviewer 3 Report

This review analyses the possible role of the elongation processes of both the mesocotyl and the primary root of maize seedlings in the responses to adverse environmental conditions. In this context, the environmental signals, such as dark and light as well as the possible molecular mechanisms involved in the control of elongation processes of these two organs are considered. The authors, starting from a short description of the anatomical and physiological aspects, presented the regulatory mechanisms and therefore the function of root and mesocotyl elongation. In many cases the quality of the review did not resulted satisfactory. The different aspects are mixed with each other as well many concepts are many concepts are repeated several times. The quality of the figures must be improved from both the graphical and the conceptual point of view. Moreover, some Figures are not correctly contextualized with the text. The legends of the figure must be improved.

Figures 1 and 2: the quality of the figure is poor from the graphical point of view. The size of the lettering is too small.

Figure 3: The figure is not clear. The authors must explain better the different phases as well as the meaning of the blue and yellow lines. Some abbreviations reported in the legend are not present in the figure. The size of the lettering is too small.

The Figure 4 is cited only at the last of paragraph 3.1 Light (line 231). This paragraph does not consider/explain many of the aspect summarized in the figure 4. The authors must improve this part as well as to regard it also in the paragraph 3.2 Hormones.

Paragraph 3.2 resulted very shallow as well as do not consider many molecular aspects regarding hormone metabolism. The authors must rewrite this part.

The paragraph 5. Deep Planting must be moved before of paragraph 3. It, in fact, introduces some very preliminary information. In this position it appears quite redundant because some concepts are already previously considered.

Lines 427-443. This part is an introduction. The authors must move it in the first part of the Review.

In the paragraph 6 the QTL, GWAS and RNA-seq studies are summarized. This section must be improved. Now, it is not informative and quite confused. In other words, the reader cannot understand their real impact on the topic of this review.

In the light of all changes, the conclusions must be rewritten.

Author Response

Dear Reviewer 3

Thank you! We found your comments extremely helpful and have revised them accordingly.

#Manuscript ID: plants-1198014 Type of manuscript: Review Title: Primary root and mesocotyl elongation in maize seedlings: Two organs with antagonistic growth below the soil surface.

  • Comment 1: Figures 1 and 2: the quality of the figure is poor from the graphical point of view. The size of the lettering is too small.

Response: We totally agreed with your comment, the edition of the images was not satisfactory for us either. Before uploading the review, it seemed to meet now the quality and size of the lettering.

Changes: We improved the quality image and increase the size font. These figures change numbers for 2 and 3 respectively.

  • Comment 2: Figure 3: The figure is not clear. The authors must explain better the different phases as well as the meaning of the blue and yellow lines. Some abbreviations reported in the legend are not present in the figure. The size of the lettering is too small.

Response: We agreed with your observation and we improved the marked points.

Changes: We improved the quality of the images and increase the size font. In addition, we try to clarify the purpose of this figure and we explained in detail the blue and yellow lines in the legend. Figure 3 changed number to Figure 4. The additional abbreviatures in the legend were informative, but we decided to remove them in order to avoid confusion.

  • Comment 3: The Figure 4 is cited only at the last of paragraph 3.1 Light (line 231). This paragraph does not consider/explain many of the aspect summarized in the figure 4. The authors must improve this part as well as to regard it also in the paragraph 3.2 Hormones.

Response: We made a terrible mistake in this case, thank you very much for indicating it. This Figure 4 is in the revised text Figure 5.

Changes: We change the position of the Figure to the paragraph 3.2.2 Hormones, where it is considered.

  • Comment 4: Paragraph 3.2 resulted very shallow as well as do not consider many molecular aspects regarding hormone metabolism. The authors must rewrite this part.

Response: We decided to complement the information in this paragraph and deepen it. However, the actors and the molecular mechanisms on our traits of interest are still unclear. This is one of the goals of our current investigation.

Changes: We added new information on hormones, and the paragraph 3.2; this is 3.2.2 in           the revised text.

  • Comment 5: The paragraph 5. Deep Planting must be moved before of paragraph 3. It, in fact, introduces some very preliminary information. In this position it appears quite redundant because some concepts are already previously considered.

Response: We entirely approved you observation.

Changes: We moved paragraph 5 before paragraph 3 for introducing the concept of deep            planting in advance and for avoiding being redundant.

  • Comment 6: Lines 427-443. This part is an introduction. The authors must move it in the first part of the Review.

Response: We gladly considered your observation.

Changes: We changed the position of these particular lines to the introduction.

  • Comment 7: In the paragraph 6 the QTL, GWAS and RNA-seq studies are summarized. This section must be improved. Now, it is not informative and quite confused. In other words, the reader cannot understand their real impact on the topic of this review.

Response: We tried to explain this section more thoroughly and its impact on our research subject.

Changes: We have separated the subject and have given them individual numbering for expressing our ideas more clearly, and we have also rewritten some statements.

  • Comment 8: In the light of all changes, the conclusions must be rewritten.

Response: We added one more conclusion in relation with your suggestion hoping that met your expectations.

Changes: The conclusions are:

                  -Deep planting should be a more common practice for developing sustainable agriculture      in arid and semi-arid areas of the Earth in the current climate crisis due to human activity.

            -New genetic and molecular tools should be used to examine the role of hormones and           their interplay in controlling root hydrotropism and mesocotyl elongation. This           characterization in maize will provide many potential targets for agronomic improvement.

            -Genetic diversity of native maize landraces should be exploited for the selection of those       who showed a robust hydrotropic response and higher elongation rate of the mesocotyl for      deep planting in poor areas without high input agriculture and prone to longer droughts       and higher temperatures.

            -Maize improvement by genetic selection should use the information derived of the    genomic and transcriptomic analysis of root hydrotropism and mesocotyl elongation traits           during seedling establishment in relation to early vigor.

Round 2

Reviewer 3 Report

According to the suggestions/comments, the authors have improved the manuscript. This new version appears quite informative, and it could be useful for further researches in this field.